# A Novel System for Semiautomatic Sample Processing in Chronic Myeloid Leukaemia: Increasing Throughput without Impacting on Molecular Monitoring at Time of SARS-CoV-2 Pandemic

**DOI:** 10.3390/diagnostics11081502

**Published:** 2021-08-20

**Authors:** Stefania Stella, Silvia Rita Vitale, Michele Massimino, Adriana Puma, Cristina Tomarchio, Maria Stella Pennisi, Elena Tirrò, Chiara Romano, Federica Martorana, Fabio Stagno, Francesco Di Raimondo, Livia Manzella

**Affiliations:** 1Department of Clinical and Experimental Medicine, University of Catania, 95123 Catania, Italy; silviarita.vitale@gmail.com (S.R.V.); michedot@yahoo.it (M.M.); adry.p88@hotmail.it (A.P.); cristina.tomarchio@hotmail.it (C.T.); perny76@gmail.com (M.S.P.); chiararomano83@gmail.com (C.R.); fede.marto.fm@gmail.com (F.M.); manzella@unict.it (L.M.); 2Center of Experimental Oncology and Hematology, A.O.U. Policlinico “G.Rodolico—San Marco”, 95123 Catania, Italy; elena.tirro@unipa.it; 3Department of Surgical, Oncological and Stomatological Sciences, University of Palermo, 90121 Palermo, Italy; 4Division of Hematology and Bone Marrow Transplant, A.O.U. Policlinico “G.Rodolico—San Marco”, 95123 Catania, Italy; fsematol@tiscali.it (F.S.); diraimon@unict.it (F.D.R.); 5Hematology Department of Surgery, Medical and Surgical Specialities, University of Catania, 95123 Catania, Italy

**Keywords:** chronic myeloid leukaemia, molecular response, Q-PCR, SARS-CoV-2 infection, *BCR-ABL1/ABL1^IS^*

## Abstract

Molecular testing of the *BCR-ABL1* transcript via real-time quantitative-polymerase-chain-reaction is the most sensitive approach for monitoring the response to tyrosine-kinase-inhibitors therapy in chronic myeloid leukaemia (CML) patients. Each stage of the molecular procedure has been standardized and optimized, including the total white blood cells (WBCs) and RNA isolation methods. Here, we compare the performance of our current manual protocol to a newly semiautomatic method based on the Biomek i-5 Automated Workstations integrated with the CytoFLEX Flow Cytometer, followed by the automatic QIAsymphony system to facilitate high-throughput processing samples and reduce the hands-on time and the risk associated with SARS-CoV-2. The recovery efficiency was investigated in blood samples from 100 adults with CML. We observe a 100% of concordance between the two methods, with similar total WBCs isolated (median 1.137 × 10^6^ for manual method vs. 1.076 × 10^6^ for semiautomatic system) and a comparable quality and quantity of RNA extracted (median 103 ng/μL with manual isolation kit vs. 99.95 ng/μL with the QIAsymphony system). Moreover, by stratifying patients according to their *BCR-ABL1* transcript levels, we obtained similar *BCR-ABL1/ABL1^IS^* values and *ABL1* copies, and matched samples were assigned to the same group of molecular response. We conclude that this newly semiautomatic workflow has a performance comparable to our more laborious standard manual, which can be replaced, particularly when specimens from patients with suspected or confirmed SARS-CoV-2 infection need to be processed.

## 1. Introduction

Chronic myeloid leukaemia (CML) is a stem cell disease characterized by a unique cytogenetic marker, the Philadelphia (Ph) chromosome, arising from the reciprocal translocation between the long arm of chromosomes 9 with the *Abelson1* (*ABL1*) oncogene juxtaposed to the *breakpoint cluster region* (*BCR*) gene on chromosome 22 t(9;22) (q34;q11) [1,2,3,4]. At the molecular level, the Ph chromosome generates the *BCR-ABL1* fusion chimeric gene, encoding an oncoprotein with constitutive tyrosine kinase activity that alters the proliferation rate, survival signalling, immunological interactions, and cytoskeletal dynamics of the hematopoietic stem cell [5,6,7,8]. The Ph chromosome is detected in 95% of CML patients, in 3–5% of paediatric and 15–20% of adult acute lymphoblastic leukaemia, respectively [9,10,11,12]. In the remaining 5% of CML cases, the fusion gene is “cryptic” and located on a normal chromosome 22 or, rarely, on chromosome 9 [9,13].

In the current clinical practice, the introduction of tyrosine kinase inhibitors (TKIs) in the treatment of CML patients has generated unprecedented rates of haematological, cytogenetic and molecular responses, increasing overall survival and improving disease outcome [14,15,16,17]. The management of CML patients is defined by the current European Leukaemia Network (ELN) recommendation [18,19]. The monitoring of molecular response should be done via quantitative polymerase chain reaction (Q-PCR), which enables to stratify patients into three groups of response: (i). “optimal responders”, who will continue the same treatment; (ii). “warning individuals”, who should be monitored more carefully; (iii). “failed patients”, who need to change the current therapy due to the risk of progression and death [18,19]. Tyrosine kinase inhibitor resistance includes biological events classified as either *BCR-ABL1*-dependent and *BCR-ABL1*-independent mechanisms [20,21,22,23,24]. The identification of resistance mechanisms is critical to define new therapeutic strategy capable of selectively killing cancer cells [25,26,27]. Patients who do not reach an optimal molecular response or develop resistance can switch to another TKI [20,28,29]. On the other hand, individuals who acquire stable deep molecular response on TKIs may be eligible for treatment free-remission (TFR) [30,31,32,33].

In the last years, qualitative and quantitative molecular techniques have been developed to evaluate and measure *BCR-ABL1* transcript [4,9,34,35]. The Q-PCR has become the “gold standard” to perform serial measurements of *BCR-ABL1* mRNA and monitor patient outcomes [34,36,37,38,39]. For valid Q-PCR data, each stage of the molecular procedure should be standardized and optimized, including the quantity of blood collected, the isolation method for total white blood cells (WBCs) and RNA extraction. Particularly, the quality and quantity of WBCs isolated and total RNA are mandatory for accuracy and reproducibility of Q-PCR analyses which is crucial for clinical decision-making, such as TKIs switching or discontinuation [9,35,40]. Overall, the measurement of *BCR-ABL1* transcript by Q-PCR should be done every 3 months after TKIs therapy initiation, then at least every 3–6 months. However, this interval may vary according to the clinical context. Mainly, CML patients with TFR require monthly Q-PCR determination during the first six months after drug discontinuation, in order to determine that the desired level of molecular response is maintained over the time [19,41,42,43].

In our laboratory, *BCR-ABL1* molecular monitoring is currently performed using a manual method for total WBCs isolation and RNA extraction. However, this manual workflow is laborious, can only process small number of samples simultaneously, increase the hands-on time and the risk associated with the use of specimens from patient with suspected or confirmed severe acute respiratory syndrome coronavirus 2 (SARS-CoV-2) infection. Hence, laboratory practices aimed to increase the number of processed human blood samples and to mitigate the SARS-CoV-2-mediated biohazard risk could be required.

In this study, we compared the performance of our standard manual workflow to a newly semiautomatic method based on the Biomek i-5 Automated Workstations integrated with the CytoFLEX Flow Cytometer, followed by the automatic QIAsymphony system, to determine whether the semiautomatic workflow can reduce hands-on time, improve high-throughput sample processes, isolate comparable total RNA and obtain accurate and reproducible Q-PCR data.

## 2. Materials and Methods

### 2.1. Patient Selection

Between January 2021 and April 2021, we carried out a research study on a total of 100 adult chronic phase CML patients (Table 1). Peripheral blood (PB) samples were collected and analysed for molecular monitoring of CML at the Center of Experimental Oncology and Hematology, A.O.U. Policlinico “G. Rodolico—San Marco” of Catania. The present study matches with the Declaration of Helsinki. All participants gave written informed consent for the data to be used in this analysis.

All patients received a TKI (imatinib brand or generic, dasatinib, nilotinib or bosutinib) as a first-line treatment. Treatment response was evaluated according to the 2020 ELN criteria [19].

### 2.2. Blood Collection and White Blood Cell Isolation

One hundred CML patients provided 28 mL of PB within a single blood draw. Matched PB samples were collected in sterile 4 × 7 mL EDTA tubes (BD Vacutainer, Becton Dickinson, Franklin Lanes, NJ, USA), according to the manufacturer’s instructions. The blood samples were stored at room temperature and further processed within 24 h from blood draw. Total WBCs from matched 14 mL of PB were isolated and lysed in RLT buffer (Qiagen, Hilden, Germany). Two isolation methods were investigated (Figure 1 and Table 2), the manual (A) and the semiautomatic (B) methods.

In the manual method, red cells were removed from 14 mL of PB blood for each of 100 CML subjects by two consecutive treatments of the blood sample with red cell lysis, incubated in ice for 15 min, followed by centrifugation (7 min at 1800 rpm). Then, the entire WBCs were washed and collected in phosphate buffered saline (PBS). Cells were diluted and a count was carried out using the haemocytometer counting-chamber with a microscope. Next, 1 × 10^7^cells were lysed in RLT buffer (Qiagen, Hilden, Germany), according to the manufacturers (Figure 1). RLT lysates were stored at −80 °C until further processing.

In the semiautomatic method, we used the Biomek i-5 Automated Workstations integrated with the CytoFLEX Flow Cytometer to enables sample processing and data acquisition (Figure 1B and Table 2). Deck space of the Biomek i-5 Workstation was customized in order to isolate total WBCs from PB blood samples of CML patients (Figure 2A). EDTA tubes with identificative (ID) number of patients were first logged in a datasheet and then loaded into the Biomeck i-5 using the available samples lanes, suitable for both 3 mL and 7 mL tubes (Figure 2A, position 1 and 2). Next, blood was transferred from EDTA tubes to 50 mL tubes, located in a cooler position (Figure 2A, position 3), by an arm linked to a span-8 Pod, a system able to move in the D-Z-X axes. Cell isolations were performed according to our standard protocol, with some minor modifications. More in detail, we reduced the time of the two consecutive red cells lysis from 15 min to 10 min and introduced a third lysis step of 5 min thawing to eliminate residual red cells. Next, WBCs were harvested and collected in PBS. Both the lysis buffer and PBS were aliquoted using a gripper gifted of gripper fingers and gripper pads to grab the bark tool (Figure 2A, position 4) that is the dispenser of buffer solutions. Then, 200 µL of collected cells were dispensed in a 96-well plate (Figure 2A, position 5 and Figure 2B) and loaded into the CytoFLEX Flow Cytometer (Figure 2A, position 6) using the gripper and a plate loader, and without the need for additional robotic transport. A total of 10,000 events were counted and cells were identified using forward and side scatter. Next, the number of cells present in 1 mL of PBS was calculated by reporting the results as “cell lives events/µL (V) × 1 mL” (Figure 2B). Finally, 1 × 10^7^ cells were lysed in RLT, as above described.

### 2.3. RNA Extraction and cDNA Synthesis

Total RNA from matched samples was extracted from 1 × 10^7^ WBCs lysed in RLT buffer using two different methods: the manual method (Table 2(C)) by the RNAse Mini Kit (Qiagen, Hilden, Germany) and the automatic (Table 2(D)) protocol by QIAsymphoy technology (Qiagen, Hilden, Germany). Total RNA isolations were performed according to the manufacture’s protocol and RNA was eluted in Dnase/Rnase free water, as previously reported [44]. Purified total RNA was quantified by spectrophotometric analysis, measuring the absorbance at wavelengths of A_230_, A_260_ and A_280_ nm by the BioSpectrometer (Eppendorf, Hamburg, Germany). RNA purity was calculated by A_260/280_ ratio (~1.9–2.0) and A_260/230_ ratio (~2.0–2.2). The RNA integrity was verified by electrophoresis running samples on 1.2% denaturing agarose gels. Total RNA samples were stored at −20 °C until their use. A total of 1 µg of RNA was used to perform complementary DNA (cDNA) synthesis by using random hexamer primers (Promega, Madison, WI, USA) and moloney murine leukaemia virus reverse transcriptase (Thermo Fisher, Waltham, MA, USA), as previously reported [45].

### 2.4. Quantification of BCR-ABL1 and ABL1 Transcripts

The *BCR-ABL1* and *ABL1* transcripts were quantified using Q-PCR in the laboratory at the Centre of Experimental Oncology and Haematology, as previously reported [46], and *ABL1* was used as the reference gene for the entire group of samples. The *BCR-ABL1/ABL1* determinations were assessed according to the international scale (IS) as the ratio of *BCR-ABL1* transcript to *ABL1* transcript. The value was expressed as a percentage on the log scale and using a conversion factor calculated every year, as previously described [46]. Molecular response (MR) has been defined as previously reported [47]. In particular, MR^3^ was defined by *BCR-ABL1/ABL1^IS^* ≤ 0.1% (3-log reduction from the standardized baseline), MR^4^ was reported by *BCR-ABL1/ABL1^IS^* ≤ 0.01% (≥4-log reduction from standardized baseline), and MR^4.5^ was indicated by *BCR-ABL1/ABL1^IS^* ≤ 0.0032% (≥4.5-log reduction from standardized baseline) with >32,000 *ABL1* transcript copies [40,48]. Quantitative PCR determinations were considered of appropriate quality only in the presence of no less than 10,000 *ABL1* gene copies, as previously reported [40]. Only those subjects with *BCR-ABL1/ABL1^IS^* transcripts below 10% were included in this research study.

### 2.5. Software and Statistical Analyses

The Biomeck Software (version 5.1 9.0 (i5)) was used to process samples in the Biomek i-5 Automated Workstations. The CytExpert program (version 2.2.0.97—Beckman Coulter, Inc., Brea, California, USA) performed a count of the white blood cells in the CytoFLEX Flow Cytometer.

Differences between cells counts, *BCR-ABL1/ABL1^IS^* and *ABL1* transcript level in matched samples were calculated using Prism software v. 8.4. Statistical significance was evaluated using the Wilcoxon signed-rank test. A *p* value below 0.5 was considered statistically significant.

To evaluate the bias between the mean differences of the two methods and to estimate an agreement interval within 95% interval, a Bland–Altman plots was used.

## 3. Results

### 3.1. Patient Characteristics

Patients characteristic are summarized in Table 1. The median follow-up of the accrued population was 63 months (range 3–172). Of total patients, 57% were male and 43% were female. The median leukocyte count was 7.59 × 10^9^/L (range 3.04–21.8) and the median of haemoglobin was 12.8 g/dL (range 10.8–14.7). Thirty-eight patients showed an e13a2 (b2a2) *BCR-ABL1* rearrangement, 54 individuals presented an e14a2 (b3a2) *BCR-ABL1* fusion transcript and 8 exhibited both e13a2 and e14a2 isoforms. According to *BCR-ABL1^IS^* transcript levels, we selected the patients with a stable molecular response which were distributed in 4 groups of 25 subjects each: Group A (10% > *BCR-ABL1/ABL1^IS^* > 0.1%), Group B (0.1% > *BCR-AB1L/ABL1^IS^* > 0.01%), Group C (0.01% > *BCR-ABL1/ABL1^IS^* > 0.0032%) and Group D (*BCR-ABL1/ABL1^IS^* < 0.0032%).

### 3.2. Concordance in White Blood Cell Isolation Efficiency between the Manual and the Semi-Automatic Platform

In order to compare the WBC isolation efficiency of the newly semiautomatic system, we measured the count of white cells isolated from matched PB samples via manual and semiautomatic platforms (Table 2). We observed similar isolation efficiency between the methods, with the count results reported as cells/mL. The median cell count obtained was 1.137 × 10^6^ (range 3.23 × 10^5^–3.07 × 10^6^) with the manual method and 1.076 × 10^6^ (range 3.00 × 10^5^–2.78 × 10^6^) with the semiautomatic platform (Table 3). Interestingly, the isolation efficiencies of the two methods were not statistically different (Figure 3A). Moreover, we confirmed that counting cells on the cytoflex system provided information on samples quality compared to their matched manual-count samples. In fact, more accurate counts in the cytoflex platform were obtained observing the scatter distribution of the cell population and modifying the counting-gate (Figure 3B). Overall, we found a 100% of concordance in all patients. The Bland–Altman plot showed no consistent bias between the manual and semiautomatic methods (Figure 3C)

### 3.3. Comparison of the Performance of Manual and Automatic Extraction Methods on RNA Quality and Quantity

Next, we evaluated whether the RNA isolation method could impact on the downstream analysis for the molecular monitoring of CML patients. Therefore, we compared the quantity and quality of the RNA obtained by the column-based RNA isolation kit to the automated silica-based QIAsymphony technology. We observed that the RNA concentration measured by BioSpectrometer was comparable for both the methods in all the 100 CML patients. In detail, the median RNA concentration was 103 ng/μL (range 75–273.5) using the manual isolation kit and 99.95 ng/μL (range 75–359.4) by the QIAsymphony instrument (Table 3). No statistically significant differences were observed between the two isolation platforms. Additionally, both the RNA extraction kits isolated samples with good quality as measured by RNA spectrophotometric quantification at wavelengths of A_260/280_ (median value: 1.9 manual vs. 1.9 automatic) and A_260/230_ (median value: 2.1 manual vs. 2.1 automatic).

### 3.4. Concordance of Quantitative PCR Performance According to the Manual and the Semiautomatic Platform

Consolidated data has established that the correct interpretation of serial *BCR-ABL1/ABL1^IS^* values depends on the accuracy of the used method [9,35]. Variables of different procedure may affect the reliability of individual assays, interfere with the measurement of *BCR-ABL1/ABL1^IS^* transcript and produce different values for scoring molecular response. Therefore, we compared the Q-PCR data obtained by the manual and the semiautomated platforms to investigate whether similar *BCR-ABL1/ABL1^IS^* values can be reliably measured. To this purpose, we first stratified the CML patients according to their *BCR-ABL1/ABL1^IS^* transcript value into four groups (Group A, B, C and D) and then we compared Q-PCR measurements obtained from matched samples. Overall, measured *BCR-ABL1/ABL1^IS^* transcript levels, *BCR-ABL1* transcript copy numbers (data not shown) and ABL1 control gene copies revealed a good concordance between the two platforms. By considering the *BCR-ABL1/ABL1^IS^*, we observed that matched samples obtained a similar “IS” score and the patients were assigned to the same group of molecular response. Specifically, we observed that in Group A the *BCR-ABL1/ABL1^IS^* medians were 0.32180% (range 0.1088–7.655), for the samples performed with the manual methods and 0.42102% (range 0.1022–6.3447) for the semiautomatic platform (Figure 4A; *p* = 0.6). In Group B, the *BCR-ABL1/ABL1^IS^* medians were 0.0338% (range 0.012–0.090) using the manual technologies and 0.04588% (range 0.011–0.099) by the semiautomatic instrument (Figure 4B; *p* = 0.8). Patients recruited in Group C showed a median *BCR-ABL1/ABL1^IS^* of 0.008% (range 0.0037–0.009) when processed with the manual platform and median of 0.005% (range 0.0038–0.009) with the Biomeck i-5 system (Figure 4C; *p* = 0.1). Both methods gave a *BCR-ABL1/ABL1^IS^* undetectable for individuals in Group D (data non shown). The Bland–Altman plot showed no consistent bias between the manual and semiautomatic methods (Figure 5A–C).

Next, we evaluated the *ABL1* reference gene copies and we found that both isolation methods performed optimally with the *ABL1* gene copies measured >10,000 in all the four groups (Figure 6A–D). Moreover, we do not observe statistically significant differences in the *ABL1* gene copies of patients stratified for both the two platforms. Overall, we found a 100% of concordance. The Bland–Altman plot showed no consistent bias between the manual and semiautomatic methods (Figure 7A–D).

## 4. Discussion and Conclusions

The molecular detection of *BCR-ABL1* fusion transcript plays an essential role in the diagnosis, management and risk classification of CML patients, pervasively influencing of clinical decisions. Therefore, molecular monitoring has become of pivotal importance to document treatment responses and predict relapse [30,34,37]. To date, Q-PCR represents the “gold standard” to measure *BCR-ABL1* oncogene transcript levels and to monitor the kinetics of disease burden reduction. Upon TKI therapy initiation, Q-PCR should be repeated every 3 months until MR^3^ is achieved, and then continued every 3–6 months [49,50,51]. Moreover, a group of CML patients with persistent deep molecular response after TKIs discontinuation (a condition defined as TFR) should be monitored more strictly to ensure a timely recognition of relapse [31,41,52].

Translating this dense Q-PCR follow-up in the practice determines a considerable burden of work for molecular laboratories. Currently, manual separation method of total white cells from blood of CML patients is widely used. This method is often laborious and time-consuming and is hence unsuitable for high-throughput isolations. Here, we present a comparison between our standard workflow for isolation and characterization of total white blood cells from blood samples of CML patients and a new semiautomatic system, based on the Biomek i-5 Automated Workstation integrated with the CytoFLEX Flow Cytometer for cell isolation, followed by the automatic RNA QIAsymphony extractor. Although the total time to perform WBCs isolation by the semiautomatic platform is higher compared to the manual method (240 min vs. 210 min, respectively), the total time for RNA isolation is roughly similar (60 min vs. 90 min, respectively). Moreover, the semiautomatic method determined a hands-on time reduction from 210 to 30 min. for WBCs isolation, and from 90 to 15 min for RNA extraction while processing a larger number of samples (Table 2). Of interest, the numbers of samples that can be processed per run may differ according to the type of EDTA tubes used. When comparing the cost required for the different platforms, the semiautomatic method is expensive because it require extra consumables to perform WBCs and RNA isolation than manual method. However, the semiautomatic method is cost-effective, which is relevant, particularly when introducing the methodology to routine diagnostic of patients with suspected infection disease.

The Biomeck i-5 instrument is equipped with a closed cabin, which ensure that laboratory procedures remain safe for the technical staff, especially in case of specimens from patient with suspected or confirmed SARS-CoV-2 infection. Additionally, avoiding manual procedures such as samples handling on an open bench, blood transfer, solutions dispensation and white cell count, further reduce the risks deriving from the analysis of suspected viral specimens. This finding is valid not only SARS-CoV-2, but also for other virus, such as hepatitis C virus or human immunodeficiency virus which can spread by contact with infected blood.

Furthermore, the integration of the automatic Cytometer also allowed for the reduction of manual laboratory work and the facilitation of the high-throughput counting of numerous samples, even in the case of infected specimens, without interfering with the downstream analyses. Indeed, data obtained with the semiautomatic isolation methods did not differed from those retrieved with the standard manual method, both in term of quality and quantity of white blood cells. Moreover, the chance to manipulate the live cell gate on the Cytoflex software enabled a more reliable count.

Different studies have recommended the selection of an RNA extraction method able to accurately and reproducibly isolate high quality of nucleic acid minimizing the risk of degradation [9,35,36]. In our study, we also compared the RNA isolated by the column-based manual extraction kit to the magnetic-beads-based automatic QIAsymphony platform. The competing automated system enabled to increase the number of samples that can be simultaneously processed (up to 96), reduce hand-on times (from 90 to 15 min.) and preserve the quality of RNA for downstream analyses (Table 2). Of note, RNA quality and quantity were similar on the two platforms for all analysed patients.

The monitoring milestones of *BCR-ABL1* transcripts levels by Q-PCR during TKIs treatment determine whether therapy should be continued or changed [30,53]. Hence, laboratory workflow should be standardized in order to increase the sensitivity and reproducibility of *BCR-ABL1/ABL1* detection. In this contest, it is recommended that a sample have at least 10,000 *ABL1* gene copies to pass minimum quality standards. Moreover, the transcript level of the control gene is critical important to determine the kinetics of disease burden reduction and define the sensitivity of the assay independently of the oncogene *BCR-ABL1* detection. Therefore, we compared the Q-PCR data obtained from matched samples isolated with both the manual a semiautomatic platform and observed similar qualitative and quantitative results, including *BCR-ABL1/ABL1* and *ABL1* recovery efficiency. Likewise, matched samples obtained comparable “IS” score and were classified into the same group of molecular response.

Nowadays, several studies have showed that the automatic procedure is often superior to the manual method, particularly for liquid biopsy [54,55].

In conclusion, semiautomatic method satisfied all our predefined aims and displayed a similar performance compared to our standard manual method. In this regard, the semiautomatic method tested is able to reduce hands-on time, increase the number of samples processed per run, isolate high quality and quantity of RNA and obtain optimal reliability and sensitivity levels by Q-PCR. Therefore, this system can replace the more laborious manual workflow, particularly when specimens from patient with suspected or confirmed viral infection have to be processed, not only SARS-CoV-2, but also for other virus, such as hepatitis C virus or human immunodeficiency virus which can spread by contact with infected blood.

## Figures and Tables

**Figure 1 diagnostics-11-01502-f001:**
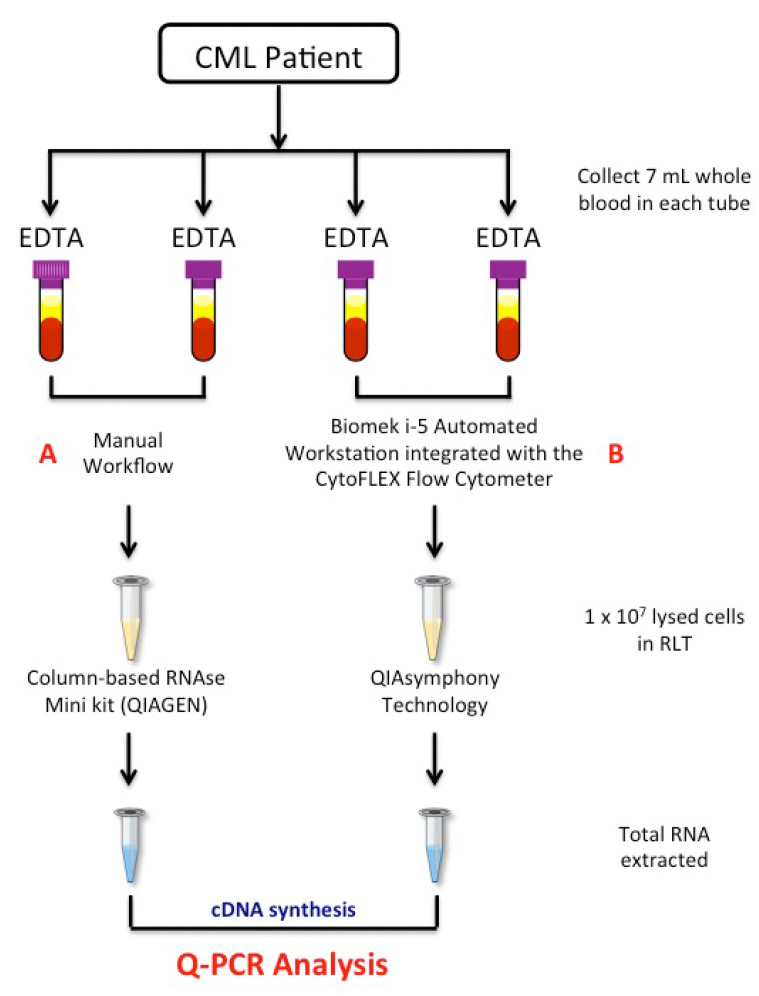
*Workflow of the study*. A total of 4 × 7 mL of peripheral blood was collected by venepuncture in EDTA tubes, with 2 × 7 mL treated by the manual (**A**) method and 2 × 7 mL processed by a newly semiautomatic (**B**) method based on the Biomek i-5 Automated Workstations integrated with the CytoFLEX Flow Cytometer. Red cells were removed from matched samples by consecutive treatments of the blood samples with red cell lysis, and 1 × 10^7^ of the collected WBCs cells were lysed in RLT buffer. Total RNA was extracted by the manual column-based RNAse Mini Kit or the automatic QIAsymphony extractor system. Quantitative polymerase chain reaction was used to measure *BCR-ABL1*, *ABL1* and *BCR-ABL1/ABL1^IS^* gene transcripts levels. EDTA tube: EthylenDiaminoTetracetyc Acid tube; RNA: RiboNucleic Acid; Q-PCR: quantitative polymerase chain reaction; cDNA: complementary DeossiNucleic Acid.

**Figure 2 diagnostics-11-01502-f002:**
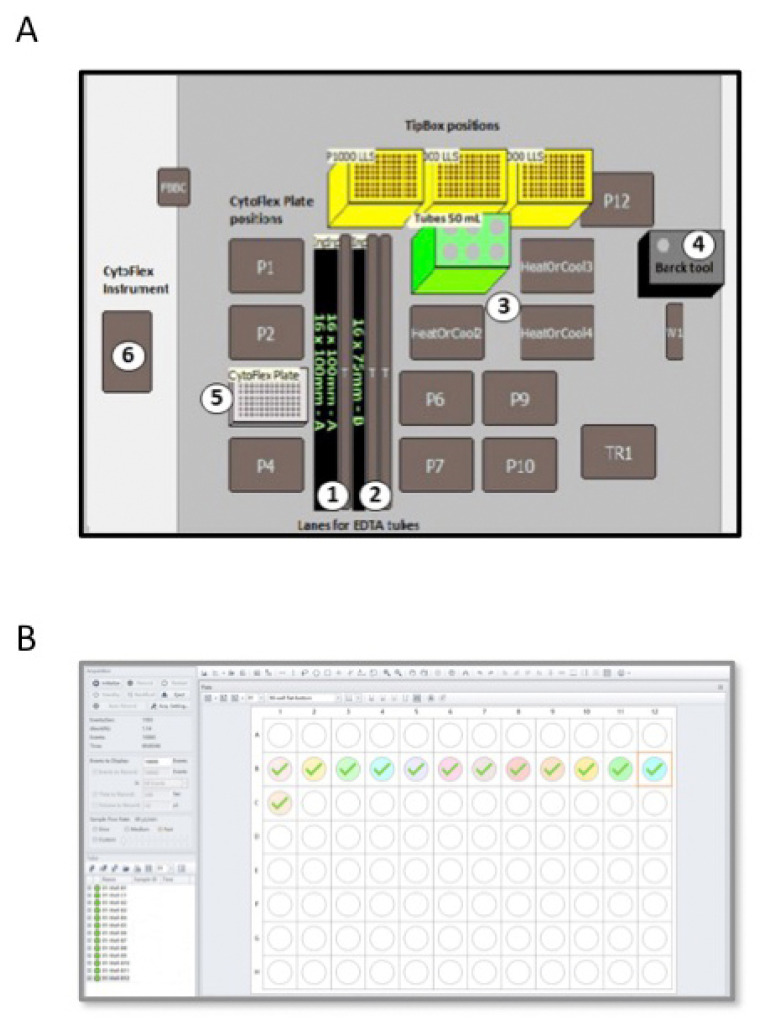
*Scheme of the Biomek i-5 Automated Workstations **(A)** integrated with the CytoFLEX Flow Cytometer* (**B**). (**A**) The figure shows the deck space of the Biomek i-5 Workstation customized in order to isolate total WBCs from PB of CML patients. EDTA tubes (3 mL or 7 mL) are loaded in samples lanes (positions 1 and 2) and blood is transferred to 50 mL tubes placed in the 50 mL tube position (position 3) by an arm inked to a span-8 Pod and using P1000 tips. The dispenser bark tool (position 4) aliquots the lysis buffer and PBS solution. Collected white blood cells are dispensed in a 96-well plate located at the Cytoflex plate position (position 5). (**B**) The figure depicts a plate Settings window of the CytoFLEX Flow Cytometer employed in the software for cell count analysis. The software allowed selection of the desired acquisition settings and the channels to set compensation and select element to record, including time and volume. Sample names are labelled in the software representing a 96-well plate. The well position on the plate matches the well position selected in the software. A total of 10,000 events are counted and cells are identified using forward and side scatter.

**Figure 3 diagnostics-11-01502-f003:**
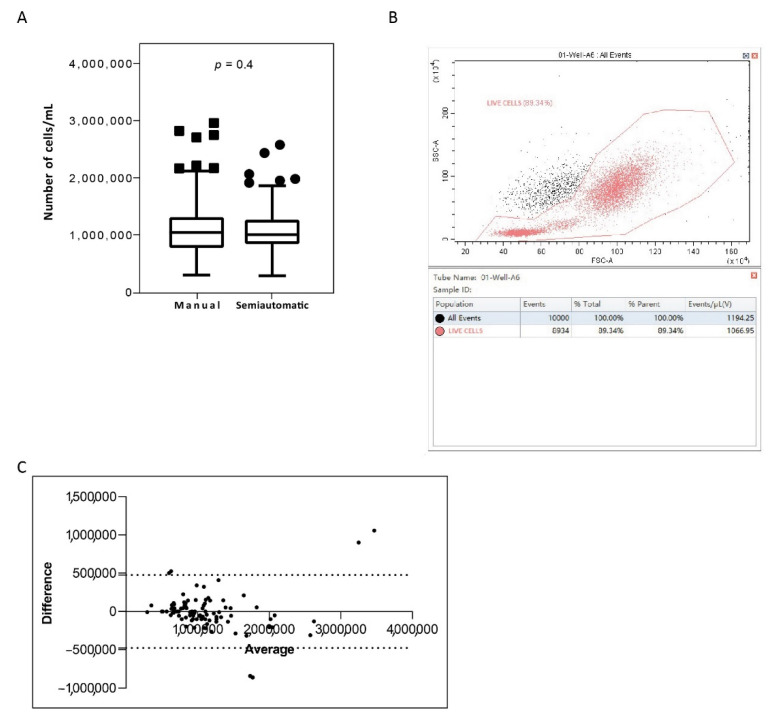
*White blood cell isolation efficiency on blood samples processed with the manual and semiautomatic protocols.* (**A**) White blood cells were isolated from 14 mL matched peripheral blood samples of CML patients (N = 100) with a manual method or a newly semiautomatic method based on the Biomek i-5 Automated Workstations integrated with the CytoFLEX Flow Cytometer. Cell counts, expressed as number of cells/mL, were analysed to determine cell recovery efficiency. The number of cells was determined for each method and depicted as boxplots delimited by the 25th (lower) and 75th (upper) percentile. Horizontal lines above and below each boxplot indicate the 5th and 95th percentile, respectively. Thick lines in each boxplot represent number cell median/mL in each method. The Wilcoxon signed-rank test was used to test the difference between the platforms. The symbols ■ and ● indicate the manual and the semiautomatic method, respectively. *p* value below 0.5 was considered statistical significant. (**B**) The figure shows the acquisition screen on the CytoFLEX Platform. The CytExpert software includes gates to visualize the distribution of total white cells isolated from blood of CML patient. Cell lives are indicated by the gate and depicted in rose. SSC-A: Side scatter-area; FSC-A: forward scatter-area. (**C**) The graph is plotted on the XY axis where X represents the difference of the two measurements, and the *Y*-axis shows the mean of the two measurements. Horizontal lines are drawn at the mean difference between the two methods and the upper and lower limits of agreement. The 95% confidence intervals are shown for the mean and the upper and lower limits of agreement.

**Figure 4 diagnostics-11-01502-f004:**
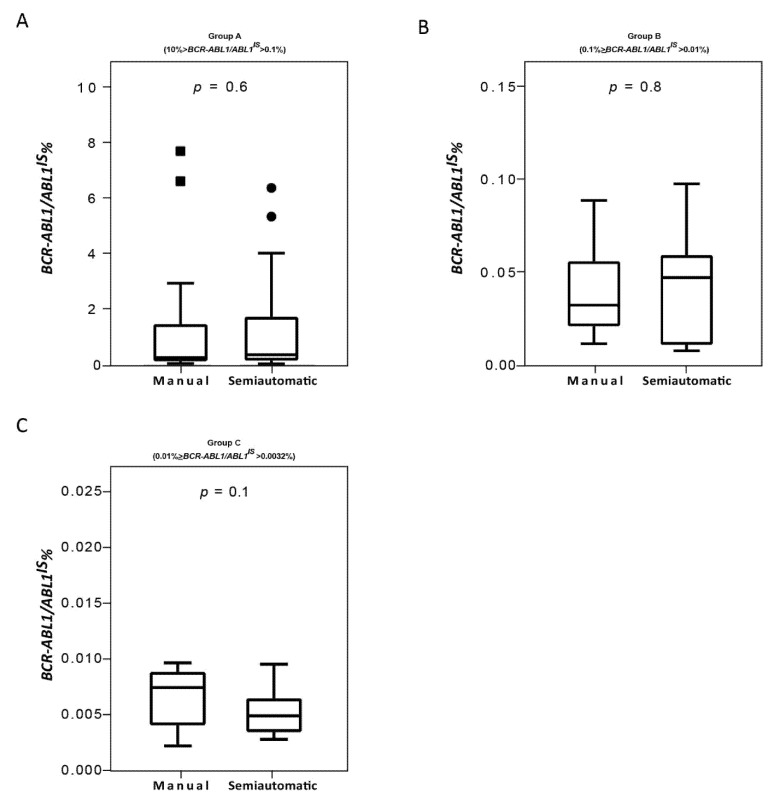
*Quantitative PCR performance on samples processed with the manual and semiautomatic protocols*. Molecular measurement of *BCR-ABL1* transcripts levels by Q-PCR of matched blood samples processed with a manual or a newly semiautomatic method. *BCR-ABL1/ABL1^IS^* had assessed in patients stratified in four groups, each consisting of 25 subjects: Group A (10% > *BCR-ABL1/ABL1^IS^* > 0.1%) (**A**), Group B (0.1% > *BCR-ABL1/ABL1^IS^* > 0.01%) (**B**), Group C (0.01% > *BCR-ABL1/ABL1^IS^* > 0.0032%) (**C**). *BCR-ABL1/ABL1^IS^* levels were determined for each method and depicted as boxplots delimited by the 25th (lower) and 75th (upper) percentile. Horizontal lines above and below each boxplot indicate the 5th and 95th percentile, respectively. Thick lines in each boxplot represent median *BCR-ABL1/ABL1^IS^* in each patient group. The Wilcoxon signed-rank test was used to test thedifference between the platforms. The symbols ■ and ● indicate the manual and the semiautomatic method, respectively. A *p* value below 0.5 was considered statistical significant.

**Figure 5 diagnostics-11-01502-f005:**
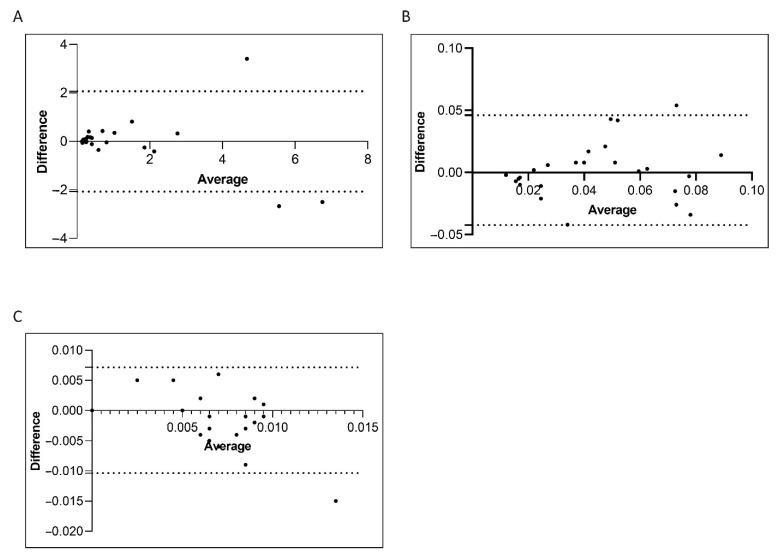
*Bland–Altman showing the concordance of the BCR-ABL1/ABL1^IS^ transcript level measured by manual and semiautomatic methods*. Paired measurements of *BCR-ABL1/ABL1^IS^* were combined for patients stratified in four groups, each consisting of 25 subjects: Group A (10% > *BCR-ABL1/ABL1^IS^* > 0.1%) (**A**), Group B (0.1% > *BCR-ABL1/ABL1^IS^* > 0.01%) (**B**), Group C (0.01% > *BCR-ABL1/ABL1^IS^* > 0.0032%) (**C**). The graph is plotted on the XY axis where X represents the difference of the two measurements, and the *Y*-axis shows the mean of the two measurements. Horizontal lines are drawn at the mean difference between the two methods and the upper and lower limits of agreement. The 95% confidence intervals are shown for the mean and the upper and lower limits of agreement.

**Figure 6 diagnostics-11-01502-f006:**
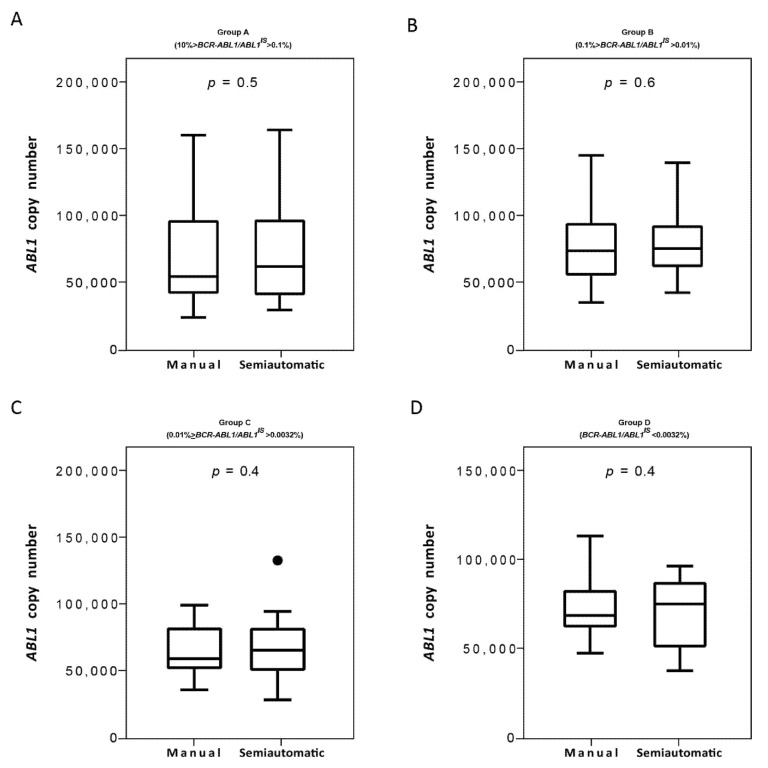
*Measurement of ABL1 control gene on samples processed with the manual and semiautomatic protocols*. Molecular measurement of *ABL1* transcripts levels by Q-PCR of matched blood samples processed with a manual or a newly semiautomatic method. *ABL1* reference gene copies were measured the four groups of patients stratified, each consisting of 25 subjects: Group A (10% > *BCR-ABL1/ABL1^IS^* > 0.1%) (**A**), Group B (0.1% > *BCR-ABL1/ABL1^IS^* > 0.01%) (**B**), Group C (0.01% > *BCR-ABL1/ABL1^IS^* > 0.0032%) (**C**) and Group D (*BCR-ABL1/ABL1^IS^* < 0.0032%) (**D**). *ABL1* levels were determined for each method and depicted as boxplots delimited by the 25th (lower) and 75th (upper) percentile. Horizontal lines above and below each boxplot indicate the 5th and 95th percentile, respectively. Thick lines in each boxplot represent median *ABL1* in each patient group. The Wilcoxon signed-rank test was used to test the difference between the platforms. The symbol ● indicate the manual and the semiautomatic method, respectively. *p* value below 0.5 was considered statistical significant.

**Figure 7 diagnostics-11-01502-f007:**
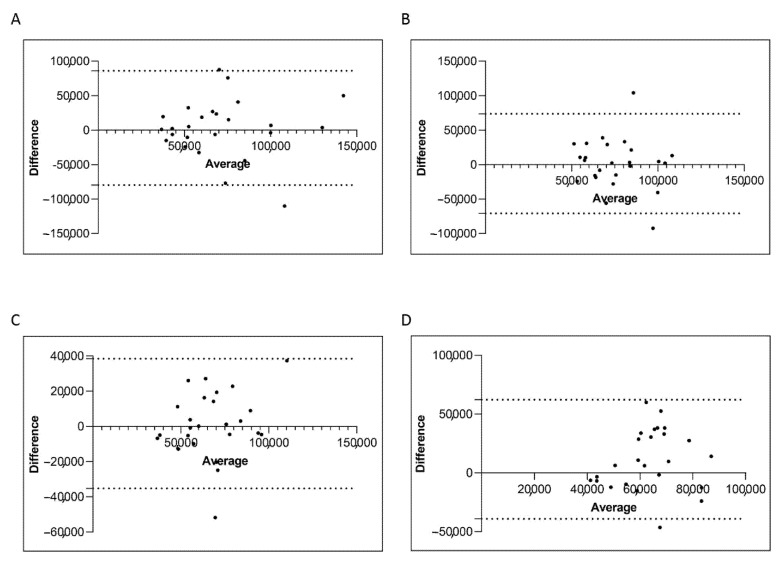
*Bland–Altman showing the concordance of the ABL1 level measured by manual and semiautomatic methods.* Paired measurements of *ABL1* were combined for patients stratified in four groups, each consisting of 25 subjects: Group A (10% > *BCR-ABL1/ABL1^IS^* > 0.1%) (**A**), Group B (0.1% > *BCR-ABL1/ABL1^IS^* > 0.01%) (**B**), Group C (0.01% > *BCR-ABL1/ABL1^IS^* > 0.0032%) (**C**) and Group D (*BCR-ABL1/ABL1^IS^* < 0.0032%) (**D**). The graph is plotted on the XY axis where X represents the difference of the two measurements, and the *Y*-axis shows the mean of the two measurements. Horizontal lines are drawn at the mean difference between the two methods and the upper and lower limits of agreement. The 95% confidence intervals are shown for the mean and the upper and lower limits of agreement.

**Table 1 diagnostics-11-01502-t001:** Patient Characteristics (N = 100).

Characteristics	N.
**Follow up**	
Median (mo.)	63
Range	3–172
**Sex (pts n.)**	
Male	57
Female	43
**Leukocyte (×10^9^/L)**	
Median	7.59
Range	3.04–21.8
**Platelet count (×10^9^/L)**	
Median	319
Range	82–740
**Haemoglobin (g/dL)**	
Median	12.8
Range	10.8–14.7
**Transcript Type**	
e13a2 (b2a2)	38
e14a2 (b3a2)	54
e13a2 and e14a2	8
**Molecular response**	
**GROUP A** (10% > *BCR-ABL1/ABL1^IS^* > 0.1%)	25
**GROUP B** (0.1% ≥ *BCR-ABL1/ABL1^IS^* > 0.01%)	25
**GROUP C** (0.01% ≥ *BCR-ABL1/ABL1^IS^* > 0.0032%)	25
**GROUP D** (*BCR-ABL1/ABL1^IS^* ≤ 0.0032%)	25

**Table 2 diagnostics-11-01502-t002:** Description of manual and semiautomatic protocols.

	Protocol	Instrument or Manual Kit	Sample	Input (mL)	Number of Samples per Run	Handling-Time per Run (min)	Cost (€) per Sample
**WBCs** **isolation**	(**A**) Manual	-	Blood	14	12	210	10
	(**B**) Semi-automatic	Biomeck i5 *&* Cytoflex		14	48 **	30	20
**RNA** **isolation**	(**C**) Manual	RNeasy Mini kit (QIAGEN)	WBCs *	0.6	12	90	8
	(**D**) Automatic	QIAsymphony (QS)		0.6	24 **	15	17

* WBCs are resuspended in RLT Buffer; ** upon request, the number of samples per run can be increase up to 96. WBCs: white blood cells, RNA: RiboNucleic Acid; min = minutes.

**Table 3 diagnostics-11-01502-t003:** Specification of white blood cell sand RNA isolation methods.

	Cells Isolation		RNA Isolation		
Protocol	Cells/mL Median Range	Total Cells Median Range	ng/µL Median Range	260/280 Median Range	260/230 Median Range
Manual	1.137 × 10^6^ (3.23 × 10^5^–3.075 × 10^6^)	5.60 × 10^7^ (1.6 × 10^7^–1.53 × 10^8^)	103.00 (75–273.5)	1.9 (1.9–2.0)	2.1 (2.0–2.2)
Semiautomatic	1.076 × 10^6^ (3.00 × 10^5^–2.78 × 10^6^)	5.3 × 10^7^ (1.5 × 10^7^–1.4 × 10^8^)	99.95 (75–359.4)	1.9 (1.90–2.0)	2.1 (2.0–2.2)

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
