# Peer review of "A Novel System for Semiautomatic Sample Processing in Chronic Myeloid Leukaemia: Increasing Throughput without Impacting on Molecular Monitoring at Time of SARS-CoV-2 Pandemic"

_diagnostics, 2021, doi:10.3390/diagnostics11081502_

Round 1
Reviewer 1 Report
Authors described about automatic sample procedure is useful as well as manual procedure.
- How much total time to perform automatic procedure or manual procedure? These are different?
- Automatic procedure is cost effective?
- Automatic procedure is often superior than manual procedure. Please, cite the article(Oncotarget 2018, 18:9: 25181-25192)
Author Response
Author’s Replay to the Reviewer 1:
- How much total time to perform automatic procedure or manual procedure? These are different?
Authors Response:
We modified the text of the Discussion to explain the total time to perform the two procedures.
- Automatic procedure is cost effective?
Authors Response:
We modified the text of the Discussion to explain the cost effective to perform the automatic procedures.
- Automatic procedure is often superior than manual procedure. Please, cite the article (Oncotarget 2018, 18:9: 25181-25192)
Authors Response:
We modified the text of the Discussion and added the mentioned reference.
Reviewer 2 Report
The authors propose a semiautomated test to measure BCR-ABL1 to accelerate the workflow while maintaining accuracy relative to the standard manual testing. 100 CML patients were sampled, which is adequate for identifying concordance between the two methods. The number of cells/mL showed similar isolation efficiency between the methods. The patients were also classified into the same BCR-ABL1/ABL1 IS transcript groups by the novel and standard methods. This showed 100% concordance. The median IS scores were also not statistically different between the two methods.
Minor critique:
Would it be possible to create a Bland Altman plot comparing the two methods?
Author Response
Author’s Replay to the Reviewer 2:
- Would it be possible to create a Bland-Altman plot comparing the two methods?
Authors Response:
We modified the text of the Results and added the Bland-Altman plots comparing the number of cells/mL (Figure 3C), the BCR-ABL1/ABL1IS transcript levels (Figure 5A, 5B and 5C) and the ABL1 levels (Figure 7A, 7B, 7C and 7D), and obtained by the manual and the semiautomatic procedures.